# Two New Sexual *Talaromyces* Species Discovered in Estuary Soil in China

**DOI:** 10.3390/jof8010036

**Published:** 2021-12-31

**Authors:** Pei-Jie Han, Jian-Qiu Sun, Long Wang

**Affiliations:** 1State Key Laboratory of Mycology, Institute of Microbiology, Chinese Academy of Sciences, Beijing 100101, China; hanpj@im.ac.cn; 2Department of biology, School of Life Science, Shaoxing University, Shaoxing 312000, China; sunjianqiu@usx.edu.cn

**Keywords:** penicillia, polyphasic taxonomy, soil fungi, two new taxa

## Abstract

In the survey of mycobiota of mudflats in China, two new sexually reproducing *Talaromyces* sect. *Talaromyces* species were discovered and studied using a polyphasic approach. These species are named here *Talaromyces haitouensis* (ex-type AS3.160101^T^) and *Ta**laromyces zhenhaiensis* (ex-type AS3.16102^T^). Morphologically, *T. haitouensis* is distinguished by moderate growth, green-yellow gymnothecia, orange-brown mycelium, and echinulate ellipsoidal ascospores. *T. zhenhaiensis* is characterized by fast growth, absence of sporulation, cream yellow to naphthalene yellow gymnothecia and mycelium, and smooth-walled ellipsoidal ascospores with one equatorial ridge. The two novelties are further confirmed by phylogenetic analyses based on either individual sequences of *BenA*, *CaM*, *Rpb2,* and ITS1-5.8S-ITS2 or the concatenated *BenA-CaM-Rpb2* sequences.

## 1. Introduction

The estimated number of global fungal species may be in a range of 2.2 to 3.8 million, whereas the number of described species only accounts for 3–8%, which is about 120,000 [1]. The less-explored environments, for instance the mudflat areas along coastlines, are thought to be the new habitats for discovery of novel fungal taxa [2,3]. With the application of molecular techniques in the studies on the mycobiota of coastal environments, the taxonomic panorama of marine-derived fungi (facultative marine fungi) is gradually emerging. Surprisingly, the predominant fungal genera from marine environments are those that have long been considered as ubiquitously terrestrial, such as *Aspergillus*, *Penicillium*, and *Talaromyces* [2,4], whose species are considered as extraordinary producers of bioactive secondary metabolites, such as alkaloids, lactones, polyketides, etc., some of which may have great pharmaceutical prospects and many of which are new compounds [5,6].

Compared with *Aspergillus* and *Penicillium*, the species of *Talaromyces* are relatively inadequately discovered in marine environments. For example, in a list of marine fungi, there are 47 species of *Aspergillus*, 39 species of *Penicillium,* and only 6 of *Talaromyces* [7]. However, *Talaromyces* is one of the most valuable reservoirs either for species diversity or for bioactive compounds (e.g., [5,6,8]). China has the vast tidal flat areas covering ca. 2 × 10^4^ km^2^, which may possess various ecological habitats harboring new taxa of *Talaromyces* as well as the consequent bioactive compounds they produce [9].

Originally, the genus *Talaromyces* C. R. Benjam. was established to accommodate sexual species producing gymnothecial ascocarps [10], and these species had long been studied in the taxonomic schemes of *Penicillium* (e.g., [11,12]). However, the morphological, ecological, and molecular characteristics have implied the common intrinsic properties of the species in *Talaromyces* and those in subgen. *Biverticillium* Diercks of Pitt from those in subgen. *Aspergilloides*, *Furcatum,* and *Penicillium*, thus *Talaromyces* was regarded as the valid genus for these species irrespective of teleomorphs or anamorphs (e.g., [13,14,15,16,17,18,19]). In the study by Samson et al. [19], 71 species were listed in the genus *Talaromyces*. Houbraken et al. [20] introduced the genus *Rasamsonia* to accommodate thermotolerant and thermophilic *Talaromyces* species. Yilmaz et al. [21] recognized 88 species and divided *Talaromyces* into 7 sections, namely, sections *Talaromyces*, *Helici, Purpurei, Trachyspermi, Bacillispori, Subinflati*, and *Islandici*, and Sun et al. [22] subsequently described the new section *Tenues* in the genus. Up to 2021, about 175 species have been reported in *Talaromyces*, and as the largest section of the genus, sect. *Talaromyces* encompassed about 76 taxa [23,24].

In this paper, we propose two new sexual sect. *Talaromyces* species isolated from mudflats of China, namely, *T. haitouensis* sp. nov. and *T. zhenhaiensis* sp. nov., using the integration of morphological and molecular phylogenetic approaches.

## 2. Materials and Methods

### 2.1. Sampling and Fungal Isolation

Mudflat soil samples were collected from Haizhouwan Resort, Haitou Town, Lianyungang, Jiangsu Province (34°55′57″ N 119°11′56″ E, 3 m) and Zhenhai District, Ningbo, Zhejiang Province (29°58′19″ N, 121°48’19″ E, 8 m), China. Soil samples were kept in sterile centrifugal tubes (50 mL) and kept at 4 °C until used. The *Talaromyces* strains were isolated from these soil samples by using the improved dilution plating method of [25] with 0.1% agar water solution instead of water. Four distinctive strains were isolated in pure culture and have been deposited in China General Microbiological Culture Collection (CGMCC) as HR1-7 = AS3.16101; ZH3-18 = AS3.16102; WYZ25-10 = AS3.16103; NN72335 = AS3.15693.

### 2.2. Phenotypic Characterization of Strains

Colony characters were examined by inoculating the strains on the media proposed by Raper and Thom [11] (Czapek agar, Cz), Pitt [12] (Czapek yeast autolysate agar, CYA, yeast extract (Oxoid), Samson et al. [26] (5% malt extract agar, MEA, malt extract (Oxoid); Oatmeal agar, OA)), and Frisvad [27] (yeast extract sucrose agar, YES, yeast extract (Oxoid)), and incubating them at 25 °C. Colonies on CYA at 37 °C and 5 °C were also assessed. The names of colors were referenced to Ridgway [28]. To make wet mounts, a small amount of mycelium with reproductive structures was picked from the colonies on MEA and mounted in 85% lactic acid solution without staining. The microscopic characters were examined and photographed with an Axioplan2 imaging and Axiophot2 universal Microscope (Carl Zeiss Shanghai Co., Ltd., Shanghai, China).

### 2.3. DNA Extraction, Amplification, Sequencing, and Phylogenetic Analyses

Genomic DNA extraction was conducted according to Wang and Zhuang [29]. The partial β-tubulin gene (*BenA*) sequences were amplified with primers Bt2a or I2 and Bt2b [30,31], the partial calmodulin gene (*CaM*) sequences were obtained with the primers AD1, AD2 and Q1, Q2 [32]; the partial DNA-dependent RNA polymerase II second largest subunit gene (*Rpb2*) sequences were amplified with primers T1, T2 and E1, E2 [33], and the rDNA ITS1-5.8S-ITS2 (ITS) sequences were obtained with the primers ITS5 and ITS4 [34]. PCR was employed in 20 µL mixture containing 0.5 µL of each primer (10 pM/µL), 1.0 µL of genomic DNA (10 ng/µL), 10 µL of 2 × PCR MasterMix buffer (0.05 u/µL Taq polymerase, 4 mM MgCl_2_, 0.4 mM dNTPs), and 8 µL of double-distilled water (Tsingke Biotechnology Co., Ltd., Beijng, China). The amplification was conducted in an AB 2720 thermal cycler (Applied Biosystems, Drive Foster City, CA, USA) with the program consisting of 94 °C for 3 min, 30 cycles of 94 °C for 30 s, 50 °C for 30 s, and 72 °C for 30 s, and a final elongation at 72 °C for 5 min [32]. The amplicons of the anticipated length were purified and sequenced in double direction with an ABI 3700 DNA analyzer (Applied Biosystems, Waltham, MA, USA), the obtained raw sequences were proofread manually with Bioedit 5.0.9 [35], and the edited sequences were deposited in GenBank. Fifty-five strains of 49 taxa of sect. *Talaromyces* were included in the individual analysis of *BenA, CaM, Rpb2*, ITS sequences, and the concatenated *BenA-CaM-Rpb2* sequences, with *T. dendriticus* CBS 660.80^T^ of sect. *Purpurei* as the outgroup. The five sequence matrices were aligned and trimmed with “MUSCLE” implemented in MEGA version 6 to generate sequence matrices, then analyzed with the Maximum Likelihood (ML) method and subjected to 1000 bootstrap replications, with substitution model and rates among sites set as K2 + G for *BenA*, K2 + G + I for *CaM*, K2 + G + I for *Rpb2*, T92 + G + I for ITS and K2 + G + I for *BenA-CaM-Rpb2* [36], and gaps were treated as partial deletion [37]. The sequence matrices were also subjected to Bayes Inference (BI) with MrBayes 3.2 to calculate the posterior probabilities (pp) of the clades [38]; the same substitution model and rates among sites were set as above. The resulting phylograms were viewed and adjusted with MEGA version 6 and then copied and pasted to Microsoft Office PowerPoint 2010 for further processing.

## 3. Results

PCR amplification generated amplicons of *BenA* about 420 bp or 660 bp, *CaM* about 650 bp, *Rpb2* about 825 bp, and ITS about 560 bp. The trimmed alignments of *BenA, CaM, Rpb2*, ITS, and the combined *BenA-CaM-Rpb2* sequences were 371, 521, 670, 456, and 1562 characters with gaps, respectively.

The phylogenetic trees generated by either the concatenated *BenA-CaM-Rpb2* sequences or the four individual loci show three isolates (AS3.16101, AS3.16102, and AS3.15693) as two distinct species of sect. *Talaromyces* (Figure 1, Figure 2 and Appendix A). *T. haitouensis* sp. nov. (ex-type AS3.16101^T^), together with *T. aspriconidius* and *T. flavus* forms a clade with 79%/1 and 72%/1 bootstrap/pp support in *BenA-CaM-Rpb2* and *BenA* phylograms, respectively. In *CaM* phylogram, *T. haitouensis* and *T. flavus* are close related with 99%/1 bootstrap/pp support, but *T. aspriconidius* is not in the same clade with them. In *Rpb2* phylogram, *T. haitouensis* and *T. aspriconidius* form a clade but without bootstrap support and *T. flavus* forms a separate clade from them. However, the ITS phylogram does not support the close relationship of these three species and *T. haitouensis* forms a solitary clade without any relatives. *T. zhenhaiensis* sp. nov. (ex-type AS3.16102^T^) and *T. stipitatus* are consistently clustered in one clade with strong bootstrap/pp support according to *BenA-CaM-Rpb2, BenA, CaM, Rpb2,* and ITS sequences.

### Description of New Taxa

*Talaromyces haitouensis* L. Wang, sp. nov. (Figure 3).

Fungal Names: FN570868.

Etymology: named after the location where the ex-type strain was isolated.

*Holotype*: HMAS 350335 (China, Jiangsu Province, Lianyungang, Haitou Town, Haizhouwan Resort, 34°55′57″ N 119°11′56″ E, 3 m, from the riverside soil, 5 May 2021, X-Y. Liu, ex-type strain: AS3.16101 = HR1-7; GenBank: ITS = MZ045695, *BenA* = MZ054634, *CaM* = MZ054637, *Rpb2* = MZ054631).

CYA 25 °C 7 d: Colonies 22–25 mm diam, thin, radially plicate slightly, margins submerged, fimbriate; colony texture velutinous and granular due to limited gymnothecia, Pinard yellow (R. Pl. IV); sporulation absent or sparse; mycelium Mars yellow (R. Pl. III) mingled with Orange vinaceous (R. Pl. XXVII); exudate and soluble pigment absent; reverse Morocco red (R. Pl. I), Strawberry pink (R. Pl. I) at margins. MEA 25 °C 7 d: Colonies 48–51 mm diam, thin, plane, margins submerged, regular; colony texture granular due to abundant gymnothecia, Green-yellow (R. Pl. V); sporulation absent or limited, conidia en masse Olive-gray (R. Pl. LI); mycelium Green-yellow (R. Pl. V), white at margins; exudate and soluble pigment absent; reverse light Danube green (R. Pl. XXXII) centrally while Cinnamon buff (R. Pl. XXIX) at peripheries. YES 25 °C 7 d: Colonies 25–28 mm diam, thin, radially sulcate, margins on agar surface, fimbriate; colony texture velutinous and granular due to limited gymnothecia at centers, Pinard yellow (R. Pl. IV); sporulation absent or sparse, conidia en masse Light olive gray (R. Pl. LI); mycelium Pecan brown to Cacao brown (R. Pl. XXVIII), white at margins; exudate and soluble pigment absent; reverse Morocco red (R. Pl. I), Moral red (R. Pl. I) at margins. OA 25 °C 7 d: Colonies 33–35 mm diam, thin, plane, mostly submerged; colony texture slimy and sparsely granular due to limited gymnothecia in centers, Light greenish yellow (R. Pl. IV); sporulation absent; exudate and soluble pigment absent; reverse Sulphur yellow (R. Pl. V). Cz 25 °C 7 d: Colonies 14–17 mm diam, slightly thick, umbonate centrally, plane, margins submerged, fimbriate; colony texture velutinous and granular due to limited gymnothecia, Pinard yellow (R. Pl. IV); sporulation absent; mycelium Orange-vinaceous to Pale vinaceous (R. Pl. XXVII), white at margins; exudate limited, Strawberry pink (R. Pl. I); soluble pigment absent or limited, light red; reverse Nopal red (R. Pl. I). CYA 37 °C 7 d: Colonies 18–20 mm diam, slightly thick, irregularly plicate slightly, margins on agar surface, regular; sporulation absent; mycelium Shell pink (R. Pl. XXVIII); exudates and soluble pigment absent; reverse Ochraceous-salmon (R. Pl. XV). CYA 5 °C 7 d: Growth absent.

Ascomata after 14 d, globose, (400–) 450–480 μm; Asci globose, 12.5–13 μm to elliposoidal, 15 × 12–13 μm; Ascospores echinulate, broadly ellipsoidal, 5–6 × 4 μm; Conidiophores from aerial hyphae, biverticillate, stipes smooth-walled, 20–60 × 2–2.5 μm; Metulae 2–4 per stipe, 10–13 × 2–2.5 μm; Phialides 2–4 per metula, 9–12 × 2–2.5 μm; Conidia smooth-walled, pyriform to ellipsoidal, 2.5–3 × 2–2.5 μm.

Notes: *T. haitouensis* is characterized by normal growth at 25 °C and 37 °C, green-yellow gymnothecia, orange-brown mycelium, and echinulate broadly ellipsoidal ascospores.

Talaromyces zhenhaiensis L. Wang, sp. nov. (Figure 4).

Fungal Names: FN570869.

Etymology: named after the location where the ex-type strain was isolated.

*Holotype*: HMAS 350336 (China, Zhejiang Province, Ningbo, Zhenhai District, 29°58′19″ N 121°48′19″ E, 8 m, from mudflat soil, 10 September 2019, F-H Song, ex-type strain: AS3.16102 = ZH3-18; GenBank: ITS = MZ045697, *BenA* = MZ054636, *CaM* = MZ054639, *Rpb2* = MZ054633).

CYA 25 °C 7 d: Colonies 65–67 mm diam, plane, thin, margins submerged, fimbriate, umbonate centrally; colony texture velutinous, overlaid with sparse floccose mycelium; sporulation absent; mycelium Sulfur yellow (R. Pl. V) to Cartridge buff (R. Pl. XXX), white at margins; exudate and soluble pigment absent; reverse Honey yellow (R. Pl. XXX). MEA 25 °C 7 d: Colonies 43–45 mm diam, thin, plane, margins on agar surface, fimbriate; colony texture velutinous; sporulation absent, mycelium pale yellow, near Massicot yellow (R. Pl. IV), white at margins; immature gymnothecia abundant, white to Massicot yellow (R. Pl. IV); exudate and soluble pigment absent; reverse pale Orange yellow (R. Pl. XXX). YES 25 °C 7 d: Colonies 61–62 mm diam, thin, irregularly sulcate, margins on agar surface, fimbriate; colony texture velutinous with sparse floccose mycelium overlaid; sporulation absent; mycelium Cartridge buff (R. Pl. XXX) to Naphthalene yellow (R. Pl. XVI), white at margins; exudate and soluble pigment absent; reverse Raw sienna (R. Pl. III). OA 25 °C 7 d: Colonies 37–40 mm diam, thin, plane, margins on agar surface, regular; colony texture granular due to abundant gymnothecia, cream yellow to Naphthalene yellow (R. Pl. XVI); sporulation absent; mycelium white; exudate and soluble pigment absent; reverse Sulphur yellow (R. Pl. V). Cz 25 °C 7 d: Colonies 38–40 mm diam, plane, thin, margins submerged, fimbriate; colony texture velutinous; sporulation absent; mycelium pale yellow, near Martius yellow (R. Pl. IV) to Marguerite yellow (R. Pl. XXX), white at margins; immature gymnothecia abundant, white to Marguerite yellow (R. Pl. XXX); exudate and soluble pigment absent; reverse Aniline yellow (R. Pl. IV). CYA 37 °C 7 d: Colonies 56–60 mm diam, thin, plane, margins on agar surface, regular; colony texture velutinous; sporulation absent; mycelium white; exudate and soluble pigment absent; reverse Augus brown (R. Pl. III). CYA 5 °C 7 d: Growth absent.

Ascomata after 14 d, globose, 300–500 μm; Asci globose, 8–10 μm, borne in short chains; Ascospores flattened ellipsoidal, smooth-walled, with one equatorial ridge, 4–5 × 3 μm.

Additional strains examined: Hainan Province, Ledong, Mount Jianfengling, 18°42′36″ N 108°49′48″ E, 800 m, from soil, 8 November 2015, X-Z Jiang; culture AS3.15693 = NN072335; GenBank: ITS = KY007094, *BenA* = KY007110, *CaM* = KY007102, *Rpb2* = KY112592.

Notes: *T. zhenhaiensis* is characterized by fast growth at 25 °C and 37 °C, naphthalene-yellow gymnothecia and mycelium, and smooth-walled ellipsoidal ascospores with one equatorial ridge.

## 4. Discussion

Sexual sect. *Talaromyces* species such as *T. flavus*, *T. liani*, and *T. aureolinus* usually produce broadly ellipsoidal ascospores with echinulate walls, but some produce ellipsoidal ascospores with one or several ridges, such as *T. stipitatus* and *T. viridis* [21,24]. In the two novel species proposed here, ascospores of *T. haitouensis* are with echinulate walls and those of *T. zhenhaiensis* have ridged walls.

The phenotypic features, such as moderate growth, aureous ascomata and mycelium, and echinulate, broadly ellipsoidal ascospores of *T. haitouensis* are also shared by *T. flavus* and *T. aureolinus*, but *T. haitouensis* generally grows faster than *T. flavus* at 25 °C (CYA: 22–25 mm vs. 9–10 mm, MEA: 48–51mm vs. 31–32 mm, YES: 25–28 mm vs. 24–26 mm, OA: 33–35 mm vs. 30–32 mm), and in contrast to *T. aureolinus*, the new species grows well at 37 °C and shows slimy colony texture with sparse ascomata and mycelium on OA, but *T. aureolinus* does not grow at 37 °C and forms normal colonies about 33–35 mm with abundant ascomata on OA. Moreover, *T. haitouensis* produces orange-brown mycelium on CYA and YES, while *T. aureolinus* commonly produces lemon-yellow mycelium on all culture media. In micro-morphology, the initials of *T. haitouensis* consist of irregularly swollen cells, while those of *T. flavus* comprise the ascogonia encircled with antheridia, and *T. haitouensis* produces much larger gymnothecia, asci and ascospores than those of *T. flavus* (400–480 μm vs. 150–400 μm, 12.5–13 μm or 15 × 12–13 μm vs. 9.5–13.5 × 8–11.5 μm, 5–6 × 4 μm vs. 4–5.5 × 3–3.5 μm, respectively). *T. haitouensis* produces short stipes (20–60 μm), but *T. aureolinus* much longer ones (300–450 μm) from surface and substratum hyphae [11,21,24].

*T. zhenhaiensis* forms a clade with *T. stipitatus* in all phylograms inferred from individual genes and the concatenated *BenA-CaM-Rpb2* sequences with strong support (Figure 1, Figure 2 and Appendix A). The morphological differences between them are subtle, but still tangible, for instance, *T. zhenhaiensis* grows much faster than *T. stipitatus* at 25 °C (CYA: 65–67 mm vs. 32–38 mm, YES: 61–62 mm vs. 40–45 mm, OA: 37–40 mm vs. 30–35 mm) and at 37 °C (56–60 mm vs. 28–32 mm), and the color of the mycelium and gymnothecia of the new species is near cream yellow to Naphthalene yellow, which is lighter than that of *T. stipitatus* which is bright yellow with a slight greenish tint. Microscopically, though the shapes and dimensions of gymnothecia, asci, and ascospores of the two species are similar, the asci of *T. zhenhaiensis* are larger than those of *T. stipitatus* (asci: 8–10 μm vs. 6–8 μm) [11,21].

The intertidal mudflats along the coastlines consist of transitional ecological habitats from terrestrial to marine ones, which are complicated with various and variable environmental factors either chemical or physical, such as salinity, carbon, nitrogen, and phosphorus nutrients due to domestic, agricultural, and industrial sewage and wastes, anaerobic soil, and the mixture of fresh and sea water, as well as the ebb and flow of the tide. Some categories of ubiquitous fungi, for example, certain species of *Aspergillus*, *Penicillium*, and *Talaromyces* possess the robust metabolic plasticity to interact with the hash environmental factors either abiotic or biotic (e.g., [39]). Though the marine-derived strains are not distinct from their terrestrial counterparts taxonomically, the marine habitats provide the selective sampling locales to isolate certain species that are rare or difficult to be discovered from terrestrial sites, moreover, their metabolites are so diverse that they are considered as the treasure trove for discovery of new compounds of pharmaceutical interest (e.g., [40]).

## Figures and Tables

**Figure 1 jof-08-00036-f001:**
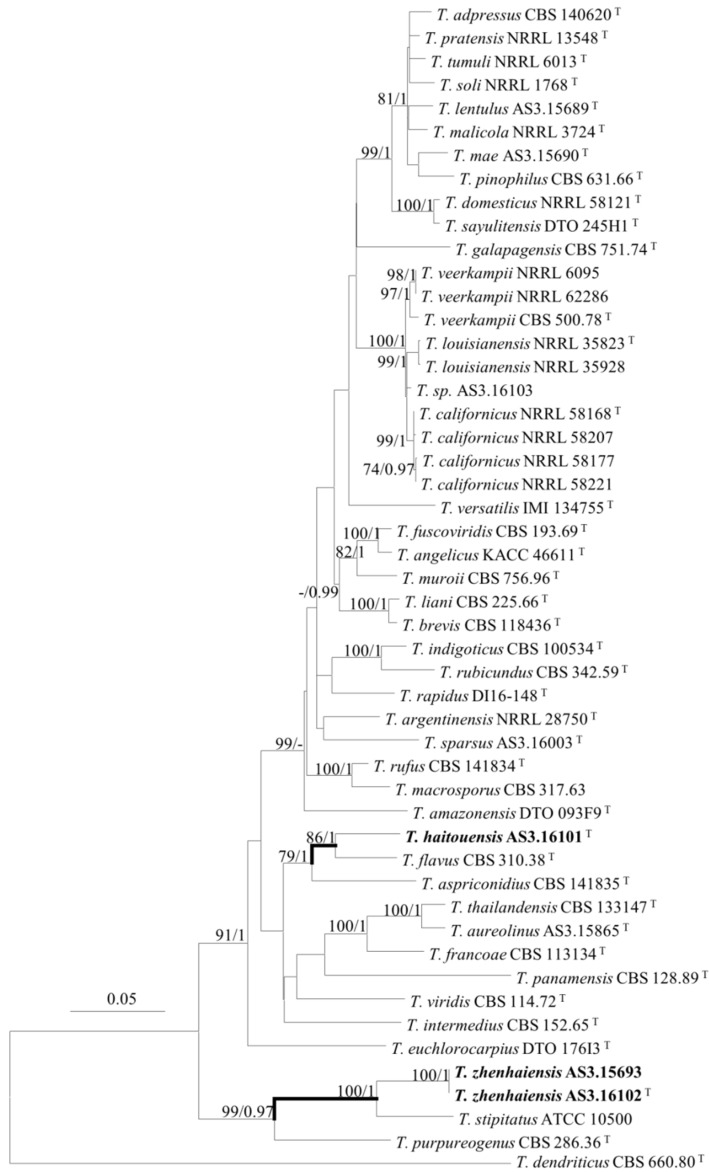
ML phylogram inferred from the concatenated *BenA-CaM-Rpb2* sequences. Bootstrap percentages over 70% derived from 1000 replicates and posterior probabilities over 0.95 of BI are indicated at the nodes. ^T^ indicates ex-type strains, strains belonging to new species are indicated in boldface. Scale Bar: number of substitutions per nucleotide position.

**Figure 2 jof-08-00036-f002:**
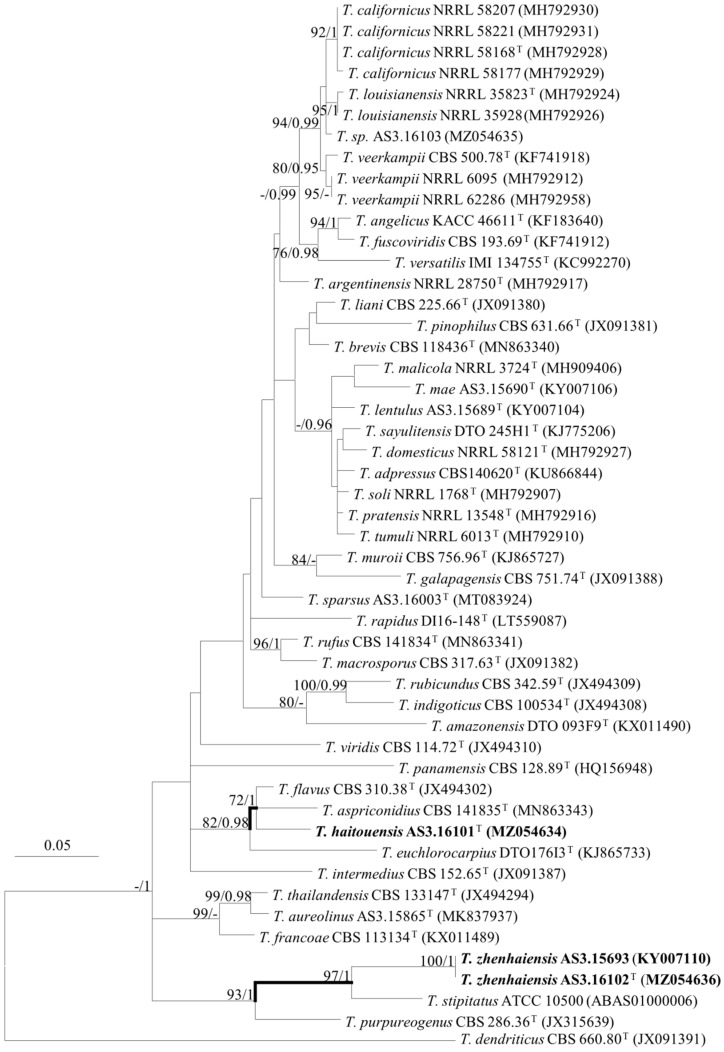
ML phylogram inferred from partial *BenA* sequences. Bootstrap percentages over 70% derived from 1000 replicates and posterior probabilities over 0.95 of BI are indicated at the nodes. ^T^ indicates ex-type strains, strains belonging to new species are indicated in boldface. Scale Bars: number of substitutions per nucleotide position.

**Figure 3 jof-08-00036-f003:**
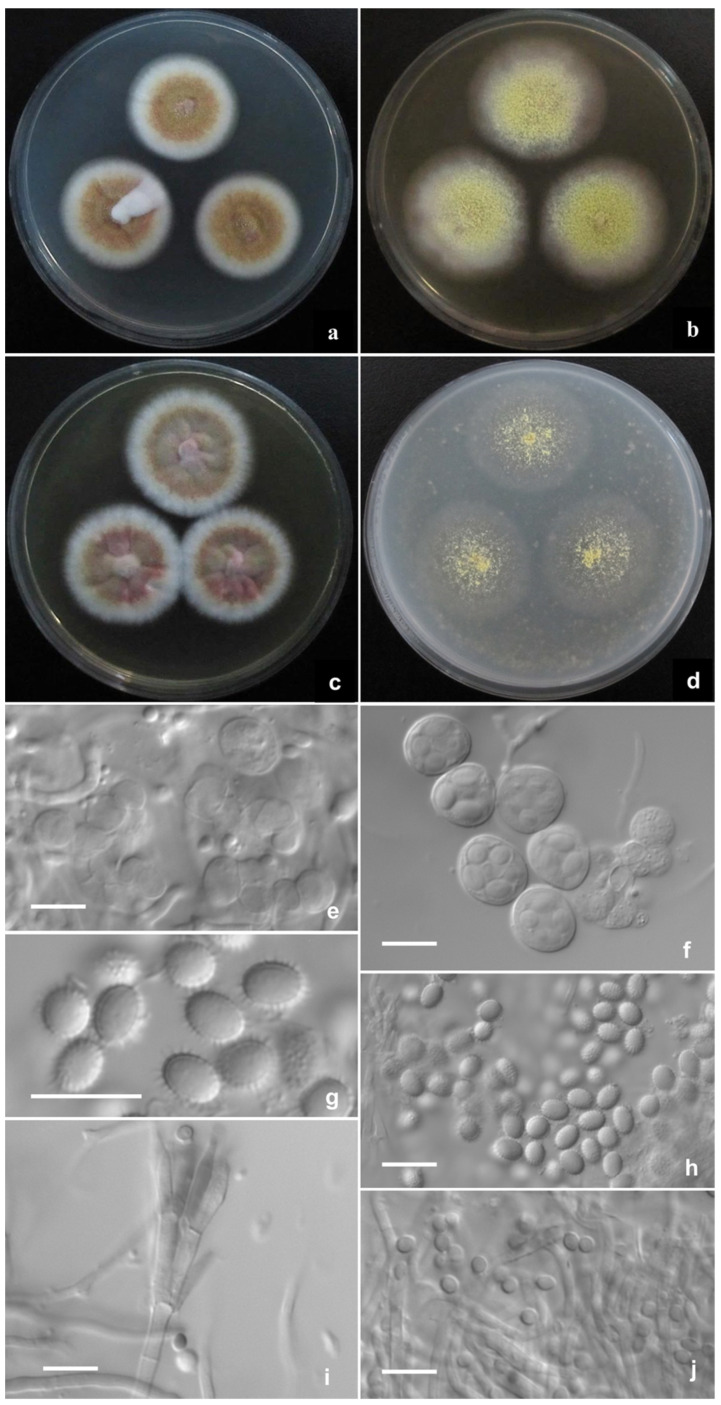
*Talaromyces haitouensis* AS3.16101^T^ incubated at 25 °C for 7 days. Colonies on (**a**) CYA. (**b**) MEA. (**c**) YES. (**d**) OA. (**e**) Initials. (**f**) Asci. (**g**,**h**) Ascospores. (**i**) Conidiophores. (**j**) Conidia. Scale Bars: 10 µm.

**Figure 4 jof-08-00036-f004:**
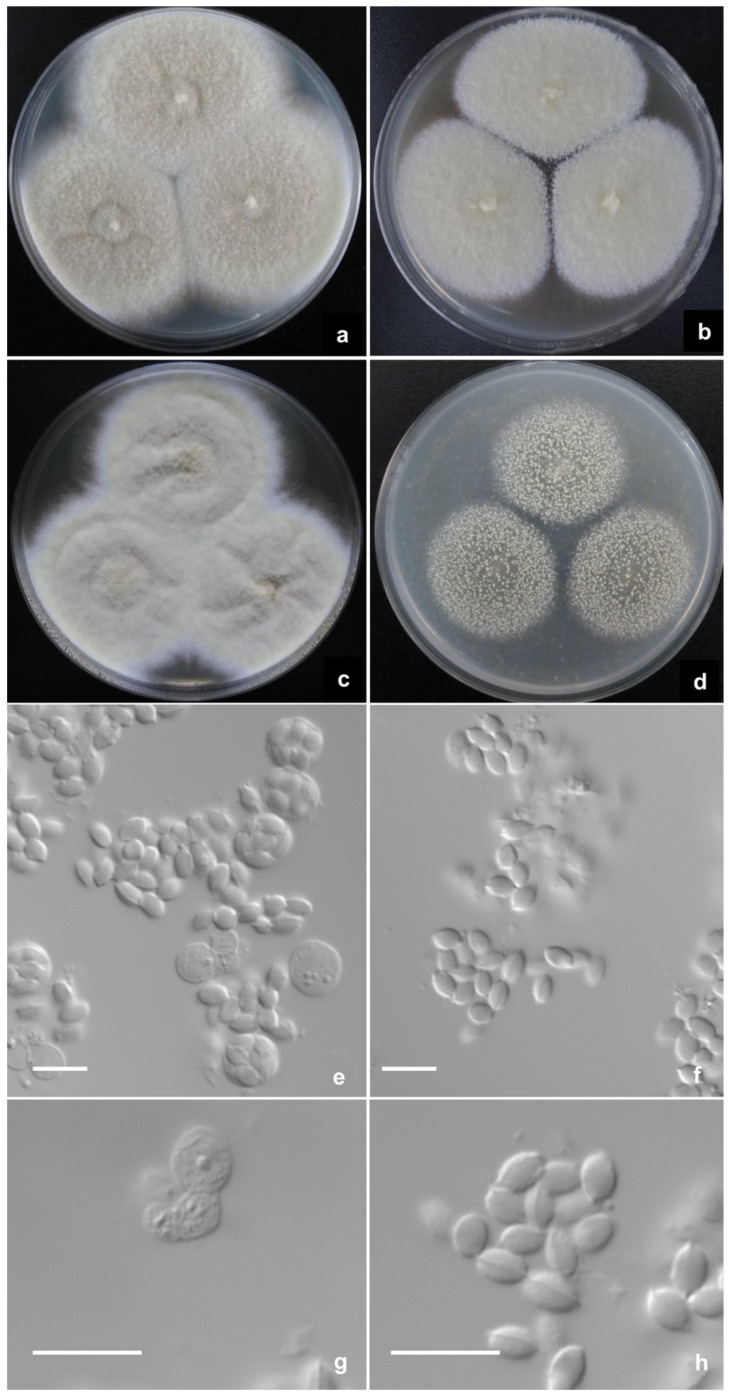
*Talaromyces zhenhaiensis* AS3.16102^T^ incubated at 25 °C for 7 days. Colonies on (**a**) CYA. (**b**) MEA. (**c**) YES. (**d**) OA. (**e**,**g**) Asci and ascospores. (**f**,**h**) Ascospores. Scale Bars: 10 µm.

## Data Availability

The sequences newly generated in this study can be found in the NCBI database.

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
