# Peer review of "Two New Sexual Talaromyces Species Discovered in Estuary Soil in China"

_jof, 2021, doi:10.3390/jof8010036_

Round 1

Reviewer 1 Report

Two novel Talaromyces  species isolated from soil in China are presented. The authors studied the macro and micromorphology of the strains and they analyzed four genetic markers. The novelty of the two taxa is adequately supported. It is an interesting study but there are some issues that need to be addressed. Detailed comments and suggestions are presented in the manuscript.

Author Response

Response to Reviewer 1 Comments

Point 1:  Two novel Talaromyces  species isolated from soil in China are presented. The authors studied the macro and micromorphology of the strains and they analyzed four genetic markers. The novelty of the two taxa is adequately supported. It is an interesting study but there are some issues that need to be addressed. Detailed comments and suggestions are presented in the manuscript.

Response 1: The detailed corrections and modification according to the comments and suggestions were marked up using the “Track Changes” function in the revised manuscript.

Some special explanations:

Point 2:Biverticillium was ranked as subgeus within genus Penicillium. The species included in this subgenus were transferred to Talaromyces”.

Response 2: In the sentence of the manuscript, we used the taxonomic scheme of Pitt (1979).

Point 3: Why CREA was not used?

Response 3: Talaromyces species generally show weak or absent growth on this medium.

Point 4: ITS has a good resolution. Why it was not included in the combined dataset?

BenA should be included in the manuscript and not as supplementary material.

Response 4: Many clades in the protein-gene phylograms are not shown in ITS phylogram.

BenA tree was added as Figure 2 in the revision.

Point 5: you mean "moderate high"?

Yes.

Point 6: It would be interesting if a concluding paragraph could be added with more general remarks regarding the novel taxa, the substrate they were isolated etc.

Conclusion remarks were added.

Reviewer 2 Report

The paper describes the discovery of two new Talaromyces species from mudflat areas in China. The paper overall is of interest to mycologists in the aspects of taxonomy. For the improvement of the manuscript, I recommend the following.

  1. The introduction is too straightforward. The authors must highlight backgrounds relevant to mycobiota of mudflats, taxonomical/biotechnological interests.
  2. The methodology lacks data as to the specific area where the samples were obtained for fungal isolation. Must inlcude GPS and sample treatments etc.

Author Response

Response to Reviewer 2 Comments

The paper describes the discovery of two new Talaromyces species from mudflat areas in China. The paper overall is of interest to mycologists in the aspects of taxonomy. For the improvement of the manuscript, I recommend the following.

Point 1: The introduction is too straightforward. The authors must highlight backgrounds relevant to mycobiota of mudflats, taxonomical/biotechnological interests.

Response 1: The introduction was enriched.

Point 2: The methodology lacks data as to the specific area where the samples were obtained for fungal isolation. Must inlcude GPS and sample treatments etc.

Response 2: The locales of sampling, the longitudes, latitudes and altitudes as well as the treatments of the samples were added to section Materials and Methods.

Round 2

Reviewer 1 Report

The manuscript has been improved substantially. There are some minor comments in the attached file.

Author Response

Response to Reviewer 1 Comments r2

Point 1: The manuscript has been improved substantially. There are some minor comments in the attached file.

Response 1: The detailed corrections and modification according to the comments were marked up by the “Track Changes” function in the revised manuscript.

Some details:

Point 2: 2.1 Which is the title of the subsections? The old version is deleted but there is no new suggested.

Response 2: 2.1 Sampling and fungal isolation

Point 3: 2.2 see previous comment

Response 3: 2.2 Phenotypic characterization of strains

Point 4: same as above

Response 4: 2.3 DNA extraction, amplification, sequencing and phylogenetic analyses

Point 5: T. aspriconidius: it seems it is a different font size

Response 5: The font size was changed into 10 points.

Point 6: Ther moderate growth, aureous ascomata and mycelium, and echinulate, broadly ellipsoidal ascospores of T. haitouensis present the similarities to T. flavus and T. aureolinusproduces

The sentence needs revising. The meaning is not evident.

Response 6: The phenotypic features, such as moderate growth, aureous ascomata and mycelium, and echinulate, broadly ellipsoidal ascospores of T. haitouensis are also shared by T. flavus and T. aureolinus,

Point 7: mycelium

Response 7: mycelium

Point 8: moreover, T. haitouensis produces orange-brown mycelium on CYA and YES, while T. aureolinus commonly produces lemon-yellow mycelium on all culture media.

It should be a separate sentence.

Response 8: Moreover, T. haitouensis produces orange-brown mycelium on CYA and YES, while T. aureolinus commonly produces lemon-yellow mycelium on all culture media.